# Microalgae-Based Biotechnology as Alternative Biofertilizers for Soil Enhancement and Carbon Footprint Reduction: Advantages and Implications

**DOI:** 10.3390/md21020093

**Published:** 2023-01-28

**Authors:** José Guadalupe Osorio-Reyes, Hiram Martin Valenzuela-Amaro, José Juan Pablo Pizaña-Aranda, Diana Ramírez-Gamboa, Edgar Ricardo Meléndez-Sánchez, Miguel E. López-Arellanes, Ma. Dolores Castañeda-Antonio, Karina G. Coronado-Apodaca, Rafael Gomes Araújo, Juan Eduardo Sosa-Hernández, Elda M. Melchor-Martínez, Hafiz M. N. Iqbal, Roberto Parra-Saldivar, Manuel Martínez-Ruiz

**Affiliations:** 1Facultad de Ciencias Biológicas, Benemérita Universidad Autónoma de Puebla, Puebla 72490, Mexico; 2Tecnologico de Monterrey, School of Engineering and Sciences, Monterrey 64849, Mexico; 3Centro de Investigaciones en Ciencias Microbiológicas del Instituto de Ciencias, Benemérita Universidad Autónoma de Puebla, Puebla 72490, Mexico; 4Tecnologico de Monterrey, Institute of Advanced Materials for Sustainable Manufacturing, Monterrey 64849, Mexico

**Keywords:** phycoremediation, phytostimulation, biofertilizer, biorefinery, carbon biocapture, circular economy

## Abstract

Due to the constant growth of the human population and anthropological activity, it has become necessary to use sustainable and affordable technologies that satisfy the current and future demand for agricultural products. Since the nutrients available to plants in the soil are limited and the need to increase the yields of the crops is desirable, the use of chemical (inorganic or NPK) fertilizers has been widespread over the last decades, causing a nutrient shortage due to their misuse and exploitation, and because of the uncontrolled use of these products, there has been a latent environmental and health problem globally. For this reason, green biotechnology based on the use of microalgae biomass is proposed as a sustainable alternative for development and use as soil improvers for crop cultivation and phytoremediation. This review explores the long-term risks of using chemical fertilizers for both human health (cancer and hypoxia) and the environment (eutrophication and erosion), as well as the potential of microalgae biomass to substitute current fertilizer using different treatments on the biomass and their application methods for the implementation on the soil; additionally, the biomass can be a source of carbon mitigation and wastewater treatment in agro-industrial processes.

## 1. Introduction

Human activities have severely impacted the environment, generating more evident and frequent climatic changes, such as heat waves, changes in marine and terrestrial ecosystems, and the loss of flora and fauna, among many others [1]. The transformation of our world and the interaction of human activities with the environment are being transformed through the implementation of the Sustainable Development Goals (SDGs) developed by the United Nations with the Member States, seeking to balance our ecosystem in terms of environmental sustainability, climate, poverty, prosperity, justice, and peace [2]. The increase in carbon dioxide (CO_2_) emissions is one of the consequences of anthropogenic activity that contributes to the increase in global warming. Microalgae are promising cell factories for the sustainable biocapture of CO_2_, and their transformation into biomass and other by-products of interest brings added value [3].

Modern agriculture needs to guarantee the needs of food and natural resources for human consumption. The intensive use of chemical fertilizers is one of the strategies to guarantee high crop yields and make the agricultural sector economically profitable. However, the intensive application of chemical fertilizers brings serious environmental problems such as: (1) soil, water, and air pollution, since only about 50% is used by plants, 2–20% evaporates through emissions of nitrogen oxides (NO, N_2_O and NO_2_), 15–25% reacts with organic compounds in the soil, and 2–10% is leached to surface water or groundwater, and (2) the degradation of crop soils derived from nutrient reduction; soil compaction; imbalance of the nitrogen (N), phosphorus (P), and potassium (K) ratio; salinization; an imbalance of the soil microbiome; and many health problems, such as intoxications and the bioaccumulation of contaminants [4,5]. Biofertilizers are an ecological alternative that allows for reducing the use of chemical fertilizers and guaranteeing the viability and use of agricultural land, as well as guarantees the sustainability of the agri-food system. Biofertilizers are friendly to the environment, improve the efficiency of use of P, N, and K and many other micronutrients, improve crop yield and quality, increase crop stress resistance, and increase beneficial interactions of the microbiome of the soil and, as a consequence, increase in protection against the toxicity of pathogens. The main biofertilizer formulations have been developed based on photosynthetic organisms, such as eukaryotic microalgae and probiotic cyanobacteria, due to their excellent results when applied to soils, mainly in increasing crop yields and improving soil fertility [6,7].

Microalgae biomass has generated a growing interest in its application in agricultural land as a powerful biofertilizer that is derived from the contribution of the high content of micro and macronutrients, bioactive compounds, and phytohormones that generate beneficial biochemical effects in the soil ecosystem derived from the interactions between crops and the soil microbiome [8]. Additionally, microalgae are microorganisms with a great capacity to recover nutrients for their biochemical processes, which allows them to grow and assimilate phosphorus and nitrogen in environments with few nutrients with high rates of specific absorption [9]. The application of live cyanobacteria and microalgae as biofertilizers induces improvements in crop growth and production yields through photosynthesis and nitrogen fixation that generates mineralization effects, mobilization of organic and inorganic nutrients, and the production of different secondary metabolites with beneficial properties, such as growth hormones, polysaccharides, antimicrobial compounds, and many others [10]. Microalgae can also be considered biocontrol agents or biopesticides derived from the potential to control or inhibit the growth of pathogens, such as fungi, bacteria, and nematodes, through the production of biocidal compounds, such as benzoic acid, and majusculonic or hydrolytic enzymes [11].

This work describes methodologies and strategies for the application of microalgae as biofertilizers and the benefits it generates in crops, soil, and all beneficial interactions with the environment in comparison with the chemical fertilizers that are currently used in intensive agricultural production as a promotion for sustainable agriculture and the circular economy of natural resources.

## 2. Chemical Fertilizer and Its Environmental Impact

Fertilizers are products that have nutrients in the form of chemical compounds, which may have an organic and/or inorganic (chemical) origin [12]. The use of chemical fertilizers dates from the mid-twentieth century to date; chemical fertilizers have helped optimize the production of food for the world population. This is thanks to the supply of nutrients, such as P, N, and K, to the crops (NPK fertilizers) [13]. In addition, some chemical fertilizers may contain the secondary micronutrients listed in Table 1 [14,15].

These products are industrially manufactured and have various commercial presentations. Fast-acting, low-cost chemical fertilizers can generally be found in solid, granular, tablet, powder, or crystalline form [15]. Inorganic fertilizers are used primarily to increase crop yields and soil fertility; however, the sustained use of inorganic fertilizers causes a decrease in soil organic matter, increases acidity, degrades the physical and structural properties of the soil, contaminates groundwater and surface water, and has carcinogenic effects on human health [24]. As a result, inorganic fertilizers have a detrimental long-term impact on the ecosystem, cause food security problems, and limit the reuse of land for agricultural production [25].

Due to their crucial role in increasing agricultural yields, fertilizers have experienced an increase of 30.8% (N), 31.4% (P_2_O_5_), and 61.3% (K_2_O), respectively, in global use. At the same time, much of these nutrients used in the fertilizer end up in wastewater, making it difficult to recover [6,26,27]. The uncontrolled use of chemical fertilizers to make agricultural production more efficient generates contamination and problems with soil reuse for agricultural purposes [25]. Researchers worldwide have pointed out that the application of chemical fertilizers adds toxic pollutants (As, Cd, and Pb) to the earth, making soils acidic, which reduces their useful life and changes the microbiota of the soil [28]. Acid soil conditions are related to the combination of fertilizers with high N_2_ content. The excessive use of nitrogenous fertilizers is the main culprit in the loss of basic cations from the soil, generating acidification and changing the biodiversity of microorganisms [29]. Microorganisms/bacteria are responsible for a variety of soil processes, such as the destruction of organic matter, nutrient cycling, and soil fertility [30]. The excessive application of fertilizers and pesticides does not always increase yields proportionally, and excessive fertilization can lead to the severe acidification of farmland, inducing metal accumulation and accelerating nutrient loss from soils, resulting in its weakening of properties and rhizodeposition [31,32]. Soil acidification due to the uncontrolled use of fertilizers has caused problems since 1970 in European countries and the USA. Furthermore, it is even on the rise in South Asia and Latin America [33]. The health and growth of plants are strongly linked to the acid and biogeochemical conditions of the soil. The effect of acid soils includes leaf damage, physiological and morphological alterations (necrotic spots and discoloration), and damage to crops and the surrounding ecosystem [34]. Certain crops are specifically sensitive to soil pH changes, such as carrots, cabbage, tomato, alfalfa, macadamia nut, avocado, and banana, and changes in pH would lead to an incomplete, damaged, or very low-quality harvest, prompting the use of more fertilizers to improve the harvest, making this an endless cycle. A study carried out [25] reports that plants use only 10–40% of chemical fertilizers; the rest remains as insoluble inorganic salts and reaches water bodies, failing the biogeochemical cycle (Figure 1).

In addition, industrial activities add to this problem with emissions of sulfuric acids (H_2_SO_4_) and nitric acid (HNO_3_) that further acidify the place and damage the ozone layer [35,36].

The nitrogenous fertilizers produced in the world generate a huge amount of reactive N_2_, which can affect food security by making the land unusable for cultivation [37,38]. Beijing, China alone is responsible for 60% of the nitrogenous compounds that reach the water by leaching in the world [39].

In addition, the urban population is still growing disproportionately, and a growth of 66% is expected by 2050, increasing people’s mobility and growth in population density, directly and indirectly; affecting surface and groundwater; and reducing its quality and potability. Contamination in water quality is a problem that requires immediate attention, even; the quality of water for agricultural activities has been classified as an environmental problem and as a topic for public policy analysis. The primary agricultural sector has been reported as the main polluter due to nitrates, phosphorus, pesticides, salt, and pathogens derived from agricultural and livestock activities [40], producing anthropogenic eutrophication, which has become a global problem because this phenomenon has the potential to affect aquatic ecosystems in the world [41]. P is the main chemical compound in eutrophication; its high concentration promotes the growth of cyanobacteria and algae, reducing dissolved oxygen in the water, as well as dissolved N_2_, which causes a decrease in the O_2_ available for aquatic fauna [42]. Water bodies saturated with P and N show a high production of algae and cyanobacteria (Figure 2), and this phenomenon can be observed in developed and developing countries; this is caused by the excessive use of chemical fertilizers [43], promoting the loss of marine biota [44].

The high concentrations of nitrates in surface waters are a problem that has been increasing in recent decades, attracting attention from different countries. A study [45] analyzed the presence of nitrates in 71 main rivers in 30 provinces of China; the results showed that about 7.83% of the samples obtained exceeded the country’s permissible limit of nitrates in water (90 mg/L), indicating severe contamination in the rivers. According to the authors, the main nitrate sources in the polluted rivers detected in the north, northwest, south, and southwest of China are positioned manure, inorganic fertilizers, and the nitrification of soil organic matter.

In other latitudes, such as North America, eutrophication problems have been growing since the 20th century due to population growth along with agricultural activities [46]. The crop of corn in the United States covers about 39 million hectares [47], and the use of fertilizers and manure is such that it is considered the main source of N_2_ in the dead zone of the Gulf of México [48]. Another example of contamination by *Microcystis* cyanobacteria in water bodies in America is what happened both in the US / Canada given by Lake [49], which reports that, since the summer of 1995, the production of cyanobacteria causes a decrease in the quality of drinking water, and this is due to anthropogenic activities that directly affect water bodies both from the use of manure and from chemical fertilizers loaded with N and P, causing the growth of cyanobacteria. Agricultural activities are the main source of contamination of bodies of water.

In the long term, agriculture can cause damage to the environment, such as soil degradation, contamination of water bodies, greenhouse gas emissions, and eutrophication. In Mexico, the northern areas of the country present this type of problem due to agricultural practices; the areas located with these problems are the Yaqui Valley, Delicias, and the lagoon region located in the states of Sonora, Chihuahua, and Durango. A study carried out by Gutiérrez et al. reports that these regions became highly productive, positioning themselves as agricultural leaders; in addition, over the years, farmers began to use chemical fertilizers and pesticides to maximize and protect their crops. However, long-term intensive agriculture increases the risk of the degradation of the soil and the environment as a consequence of the anthropogenic contamination of nitrate in water bodies, causing health problems due to the ingestion of nitrates and modifying the aquatic flora and fauna due to eutrophication [50].

The eutrophication of the waters not only favors the appearance of algae, but also the extinction of submerged plants, as has been reported in countless cases. Kemp et al. [51] studied the chronology of ecological responses to eutrophication in the Chesapeake Bay, the largest estuary in the US, showing that increased phytoplankton causes decreased water clarity and reduced dissolved O_2_. Algal growth generates severe and recurrent hypoxia in deep water, leading to the loss of several submerged vascular plants. The degradation of these benthic habitats has favored the decline of benthic macrofauna in different regions of the bay and of blue crabs in polyhaline areas. Phytoplankton growth is also associated with the accelerated decline of the bay’s endemic fish and oyster populations [51].

Another example of the consequences of eutrophication is the one exposed by Katsuki et al. [52] in Lake Makoto in Japan. The initial step of eutrophication was around the 1930s; it was gradual and increased until 1970. This phenomenon began with agricultural expansion, and after 1970, it was induced by the increase in cattle numbers and the discharge of animal feces from cattle in the Makoto River. This fecal waste in the Makoto River caused an increase in NPK nutrients, generating the proliferation of algae and cyanobacteria, reducing dissolved O_2_, and reducing the entry of sunlight at depth, killing species of Zostera marina and C. scutellum, thus causing an almost total disappearance of these species in Lake Makoto [52].

Similar degradation of freshwater ecosystems is observed around the world due to intensive agriculture in different geographic regions. The European Environment Agency concluded that only 40% of surface water bodies in Europe meet ecological standards [53].

As mentioned above the implications of nitrogen are biodiversity depletion, the effects are more notorious in species that directly obtain the nutrients by absorption. These organisms are algae, lichens, and bryophytes [54].

Few studies are focussed on fauna impact due to fertilizer uses; however, in a study carried out by Wang et al., [55] the abundance and diversity of soil under fertilization regime were evaluated. In this study, the major influence in total fauna abundance was obtained by organic fertilizer, with an increase as a result, while up to 22.6 % of diversity fauna loss was observed due to inorganic fertilizer. From the dominant groups, the abundance of *Prostigmata*, *Oribatids*, *Collembola,* and *Diptera* larvae was increased by organic and inorganic fertilizers.

Meanwhile, Wang et al. [56] report that the soil microarthropod community was evaluated against inorganic and organic fertilizers. They found that both types of fertilizers increased the abundance of microarthropods, bacterivorous Acari, and hemiedaphic and epedaphic Collembola without disturbing the taxonomic diversity of the soil. The microarthropod communities influenced the presence of invertebrates such as beetles; moreover, the presence of these communities reflected an improvement in organic matter decomposition [57]. These studies remark on the importance and influence of fauna diversity and abundance resulting after fertilizer implementation; however, as it is a complex interaction, more studies must be carried out. As described by Prashar and Shah [58], the soil microflora as well as the microbiota result are affected by the uses of fertilizers and pesticides, but remarkably, for flora and fauna abundance, the organic fertilizers or biofertilizers present more advantages than disadvantages against synthetic options.

### Impact of Chemical Fertilizers on Human Health

The prolonged use of fertilizers, as already mentioned, causes an imbalance in the soil microbiota and the acidification of it [36,40]. Additionally, because the majority of N-type fertilizers are delivered as insoluble N salts, the plant’s roots have a reduced ability to absorb nutrients, which ultimately causes leakage into bodies of water [35]. The levels of nitrate are high as groundwater is reached, especially in terrestrial areas (crops, streams, lakes, dams, etc.).

The consumption of water contaminated with nitrates or vegetables with a high nitrate content can cause serious pathological conditions [59]. The consumption of water from sources contaminated with nitrates added to the (excessive) intake of roots and vegetables generates the main source of intoxication for human health [60]. When nitrates are consumed, they have been reduced to nitrites and, therefore, to biologically active nitrogen oxides thanks to the bacterial strains located in the oral cavity, catalyzed by the enzyme nitrate reductase (NOS). After swallowing, nitrates and nitrites in the gastric acid medium are metabolized into deoxyhemoglobin, myoglobin, neuroglobin, xanthine oxidoreductase, aldehyde oxidase, carbonic oxidase, and mitochondrial enzymes and nonenzymatic pH-dependent reduction. Then, the small intestine is absorbed and the NO present in the blood ends up in the bloodstream, and the tissues can spontaneously oxidize, producing nitrites and nitrates. The excess of these compounds is excreted in the urine, and the rest is concentrated in the salivary glands [61], causing health conditions, such as methemoglobin, diabetes, cancer, and thyroid disease [40,62]. For this reason, the WHO has established an upper limit equal to 10 mg/L for drinking water [63].

The adverse effects of nitrate consumption in the human diet can be summarized in two mechanisms [59]. The first conversion route describes the formation of methemoglobin. It is a disease that originates when the degree of oxidation of the iron contained in the heme group exceeds the compensatory mechanisms of the red blood cells, passing to the ferric state, which is unable to transport oxygen and carbon dioxide. Carbon that is in high concentrations can induce hypoxia, a condition in which the body is denied adequate oxygen consumption. The second mechanism is the formation of endogenous N-nitroso compounds, the reduction of nitrate to nitrite leads to the formation of different nitrosating products in acidic stomach conditions. Nitrosating agents interact as protein amines and amides or with drug precursors resulting in the production of N-nitroso, and these compounds are potentially carcinogenic (Figure 3) [64]. It was reported that nitrosamines are related to several cancers of the digestive tract; therefore, the WHO established an upper limit of the concentration of the daily absorption of nitrate and nitrite of 3.7 mg/kg and 0.05 mg/kg of nitrate and nitrite [65].

A study by Buller et al. [66] in Iowa evaluated the dietary intake of nitrates and nitrites in women between 50–75 years old, both by the consumption of vegetables and by water with high concentrations, concerning the incidence (1986–2014) of esophageal cancers (*n* = 36), stomach (*n* = 84), small intestine (*n* = 32), liver (*n =* 31), gallbladder (*n* = 66), and bile duct (*n* = 58); in their models, they found an association between meat nitrates processed where the consumers generated some stomach cancer, just as consumption of total dietary nitrite from vegetable sources is inversely associated with gallbladder cancer; in addition, small intestine cancer was related to a high intake of animal nitrite.

In addition, the work done by Nelson et al. [67] found a positive interaction between canned vegetables as well as salty meats with high nitrate concentrations and gallbladder cancers as similar relationships for other gallbladder cancers. Biliary tract cancer cases were taken from 42 hospitals in urban Shanghai; the groups were a mix of men and women of all ages. Food frequency questionnaire data were available for 225 gallbladders, 190 extrahepatic bile ducts, and 68 blebs of Vater cancer cases.

On the other hand, research done by Calleros-Rincón et al. [68] highlights excess contamination in the bodies of water used for the supply in the town of Lerdo Durango-Mexico. The study group consisted of 103 children aged 1–12 years with high levels of methemoglobin, where children had a higher concentration of methemoglobin (due to more acidic conditions in the stomach); this may be due to contamination in the bodies of water used for human consumption, causing an increase in the concentration of nitrate in the blood and, therefore, generating oxidation in methemoglobin, which at high concentrations can generate hypoxia in the inhabitants.

## 3. Microalgae as Biofertilizers in Modern Agriculture

Because of the continued population growth, the food demand has increased, and the agricultural industry has changed, increasing the use of fertilizers to raise crop yields. In recent years, this sector has been forced to adopt an eco-friendlier approach [69]. According to Mitter et al. [70], biofertilizers are a sustainable approach to soil improvement for agriculture, and the term includes the use of single strains or a consortium of bacteria in formulations that when applied to soil they can improve its characteristics and plant growth. The viability of biofertilizers and their efficiency not only depend on the increase in crop yield, but also on the impact they can have on the soil—more specifically, the impact on the microbiological activity of the soil.

The most important element for the development of a plant is nitrogen. The content of N in the plant, alongside carbon, hydrogen, and oxygen, is the highest, and it plays an important role in the constitution of the plant structure, and it is present in chlorophyll, a pigment that has a direct impact on the photosynthesis processes, making it an essential element for the growth of the plants. The process of making this element bioavailable in the soil for plant usage is one of the main roles played by the soil microbiome: performing the fixation of atmospheric nitrogen. If the microbiome increases the content of nitrogen in the soil, the growth of the plants is maximized. Although it is considered that N_2_-fixation is only carried out by prokaryotic organisms like cyanobacteria, *Clostridium,* and *Bacillus* among others, this process can also be promoted by the addition of eukaryotic microalgae. It was shown that the application of the biomass of *Chlorella vulgaris* IPPAS C-1 with a sprinkler into bean crops showed an increased in the fixation of bioavailable nitrogen in the soil below the plant [7,71], making the use of microalgae as a natural source of N for the plants appealing.

Another chemical element important in the development of the plant is phosphorus. The P is present in the DNA and ARN bonds, and it even constitutes the molecule ATP fundamental for all the vital processes of the plant. The P that the plants require to function is normally obtained from the soil, and enriching it is one of the main goals for traditional fertilizers, but as mentioned in the last section, this can have detrimental effects on the soil and bodies of water, making it important to switch the P source for a less harmful one like the usage of microalgae. One way to solve the problem of P contamination in bodies of water and redirect this element to the soil is using microalgae to grow in said contaminated waters and applying the obtained biomass to the soil; this was carried out by Schreiber et al. [72], where he compared the effect of chemical fertilizers vs live and dry microalgae biomass on P bioavailability and the effect it can have on the growth of wheat plants. The results showed very similar plant growth between both types of fertilization with mineral fertilizer and microalgae. Although the concentration of P was lower in the plants fertilized with microalgae (both wet and dry), the similarities in root growth showed no difference between the treatments or even the positive control, demonstrating that the microalgae released N and P at a comparable rate as a traditional fertilizer.

Although biofertilizers using bacteria as the main component have been studied extensively, showing great results for a wide variety of crops [73,74], and the use of fungi-based biofertilizers is widespread too, the use of microalgae has gained strength as a versatile organism that can be used in a model of circular bio-economy in a wide range of industries from their employment for the biocapture of CO_2_ to their wastewater treatment to the production of food, energy, secondary metabolites, cosmetics, medicine, and even in the manufacturing of biofertilizers [75].

One of the benefits of using the three types of biofertilizers, apart from the plant growth promotion, is the endosymbiotic bond that the microorganisms create with the roots of the plants and the microbiome of the soil, improving the fertility by enhancing the water retention, soil structure, and the protection of the crops from pathogens and other pests, among other advantages [7]. The endosymbiotic bond is constructed by the ability of the bacteria, fungi, and microalgae-based biofertilizers to perform a variety of functions like the fixation of atmospheric nitrogen by bacteria and microalgae, the solubilization of phosphorus from these three types of microorganisms, and the release of nutrients that have shown a comparable result to the usage of traditional fertilizers, confirming the success of biofertilizers in agriculture [7,25]. Studies have shown that all the desired characteristics can be achieved with one strain of microalgae or cyanobacteria instead of using a consortium of bacteria or fungi to obtain the same results. One advantage of using microalgae-based fertilizers is decreasing greenhouse emissions by the sequestration of methane and CO_2_ while adding more organic carbon to the soil. All these benefits plus others can be achieved using microalgae as biofertilizers. However, one of the main drawbacks to their implementation on a larger scale is the lack of production on an industrial level and their commercialization, a problem that the traditional fertilizers and the biofertilizers based on bacteria or fungi do not have [25,76,77,78].

In Table 2, we present the comparison of the different types of biofertilizers commonly used against traditional fertilizers and microalgae-based biofertilizers, highlighting the advantages/disadvantages, uses, and benefits in plant growth and the environment.

### 3.1. Microalgae-Based Bofertilizer 

The use of microalgae as a soil enhancer/biofertilizer has a variety of benefits, not only for the environment but for the health of the soil and, consequently, for the crops. The percentages of moisture, pH, and light that are present when microalgae biomass is in direct contact with the soil cause the viable microalgae cells to become active and maintain their metabolic activity, aiding in the fixation of atmospheric nitrogen. This can also dose other macro and micronutrients important for the growth of the plants and can even create an endosymbiotic bond with the roots of the crops and other microorganisms present in the soil [62]. The symbiotic relationship between microalgae and plants has been studied, and Özer Uyar & Mısmıl [79] demonstrated the positive effects of growing a culture of *Chlorella vulgaris* and the mint plant *Mentha spicata* on a hydroponic system. They concluded that, among the different treatments, the co-cultivation of microalgae and plant, combined with aeration, had the greatest impact on the increase in plant weight because of the development of new shoots and leaves, and the measurement of the photosynthetic pigments on the plant revealed no stress on its growth. Another endosymbiotic bond of microalgae is with bacteria, making them thrive on different ecosystems taking advantage of one another to survive, making it a niche opportunity for the usage of this consortium in wastewater treatment, nutrient recovery, production of feedstock, biofuel, and biofertilizers [80]. It has been confirmed that the usage of microalgae and other microorganisms, even in harsh conditions like the desert, improves the fertility of the soil by enhancing its properties like water retention, maintaining its overall stability, and removing pollutants, and offering the plants a better substrate [81].

Among the benefits previously discussed, it is worth noting the important biomolecules for agriculture that can be found in microalgae; these biomolecules can improve plant productivity (biostimulants), provide plant protection against stress factors (biopesticides), and can improve soil characteristics (biofertilizers) [82]. Microalgae’s main contributions as soil enhancers are the changes in soil physical characteristics, the input of biomolecules, and the improvement in microbiological activity [83].

Microalgae-based biofertilizer contribution and effect on soils are related to the way microalgae gets into the soil and if the biomass is active or not, for example, if you are using fresh, dry, or digested biomass [7]. Based on that, the production of biofertilizers with microalgae could be relatively simple; in all cases, the first step will be microalgae biomass growth and harvest, and the differentiation will be after the harvesting, where the cells will be treated correspondingly to the type of biofertilizer formulation that is intended to have [73]. When the formulation of the biofertilizer/soil enhancer is liquid, the procedure is simple; the microalgae are grown and scaled up to an industrial level, and the obtained culture is supplemented with additives that aid to maintain the viability of the cells for longer periods [84]. On the other hand, if the biofertilizer formulation is expected to be solid, the procedure for obtaining it will require extra steps after microalgae biomass growth and harvest. These extra steps will mainly consist of the removal of water from the biomass by different techniques such as (i) lyophilization [85], (ii) air/oven drying, or (iii) carbonization [86].

#### 3.1.1. Wet Microalgae-Based Biofertilizers

The simplest and the most low-cost way of using microalgae as a biofertilizer is to use the cells in suspension or the extract of all the biomolecules contained in them; this is considered wet microalgae and can be applied from the germination state of the seeds and for the cultivation on the soil. The performance of these treatments with microalgae extracts or living cells is further explored below.

In a study, Habibi et al. [87] germinated and cultivated three varieties of rice using a suspension of cyanobacteria *Anabaena* sp., and the researchers compared the results obtained from the cultivation supplemented with the blue-green algae vs the normal treatment with water. The results showed an enhanced germination process by all three varieties of rice used, and the plant growth showed a better performance than the control, leading the researchers to recommend the use of this cyanobacteria as a viable biofertilizer.

For a more rounded approach, Mahmoud et al. [88] aimed to determine the influence of two different microalgae strains, *Chlorella vulgaris (Chlorophyta)* and *Anabaena cylindrica (Cyanobacteria)*, in the growth of *Spinacia oleracea* and to test if the microalgae supplementation, either by foliar or soil application, can decrease the accumulation of heavy metals on the spinach plant. Suspensions from both microalgae at 1 and 2% of dry weight used in both soil and foliar application were tested against spinach seeds planted onto heavy metal contaminated soils. The results showed that the microalgae suspensions promoted the growth of the spinach plant between 21 to 29% compared to the control; the best suspension occurred with *Chlorella vulgaris* at 2% and with the soil application for the growth, and it increased the macronutrients intake of N, P, and K of the plant while the treatment with *Anabaena cylindrica* at 1% concentration and foliar application performed better for a fresh and dry matter of the spinach and showed the best macronutrients intake compared to *Chlorella vulgaris*. In both cases, the microalgae reduced the heavy metal content of cadmium, lead, and copper, showing the capability of these microorganisms to accumulate these contaminants from the soil and stop them from entering the plant.

To prove the effects of microalgae as biofertilizers on different solutions, Kholssi et al. [89] studied the effect of three different solutions containing *Chlorella sorokiniana* biomass or extract of this microalgae on the growth of wheat compared to the usage of BG-11 medium as control. The three different solutions containing the extract and resuspended biomass in microalgae medium showed an increase in the germination process with the total weight of the plant and the length using solution 2, the one with only the filtered extract from the biomass, the best results of the experiment demonstrating that the extracellular biomolecules of the *Chlorella sorokiniana* are playing an important role for the cultivation of *Triticum aestivum*.

The difference in performance by dry and wet biomass compared to the usage of mineral fertilizers was tested by Schreiber et al. [72] by comparing the growth of *Triticum aestivum* by adding biomass and fertilizer to deliver between 115 to 130 mg of P per pot of plant. This experiment showed that, compared to the positive control with all the required nutrients for the growth of the plant, when the mineral fertilizer and wet and dry algae were applied to a nutrient-deficient substrate (Null Erde), the plant presented virtually the same weight in dry root and the root diameter. In the sand substrate, the best performance was observed by the mineral fertilizer, followed by the wet algae and, lastly, the dry algae in those same parameters as the previous substrate. This demonstrates that microalgae can be used as a biofertilizer with practically the same results, wet or dry; the only disadvantage is the cost of production of this biomass.

#### 3.1.2. Dry Microalgae-Based Biofertilizers

Drying microalgae biomass is the most reported method for the production and application of biofertilizers. Retrieving the biomass by letting it air-dry directly into sunlight makes it the simplest, most effective, and cheapest way of obtaining a solid biofertilizer. Other ways of drying the microalgae cells are with lyophilization; although it is an effective method, the application of this at an industrial level is highly unlikely given the elevated cost of processing per sample. Some examples of the use of dry microalgae biomass using these drying methods as a growth enhancer for a variety of crops are presented below.

An experiment using solar-dried biomass of microalgae as a fertilizer with lettuce plants was carried out [90]. It was observed that the growth rate of the lettuce showed a 121% increase compared to the control (commercial fertilizer); in addition, when the amount of ammoniacal nitrogen was similar in the commercial fertilizer and the dry biomass of microalgae, the total nitrogen was 3.5 times higher in the dry biomass of microalgae. Similarly, commercial fertilizer was also used as a control, and the biomass of the microalgae *Tetraselmis* sp. was used as a biofertilizer, applied every 2 weeks at a dose of 0.5 g. This gave the best results in the diameter of the stem, number of roots, and length of leaves in date palm plants (*Phoenix dactylefera*) compared to the control [91]. Another study was conducted comparing the growth of *Uruchloa brizantha* using a commercial chemical fertilizer and a biofertilizer made with lyophilized microalgae (mainly *C. vulgaris*) compared to a control. The results indicated that the microalgae biomass was a good option as a fertilizer, showing similar plant productivity as the one treated with the chemical fertilizer [85]. Likewise, an experiment was conducted aiming to prove the efficacy of biofertilizers in the growth of spinach and baby corn. Of the six treatments, the one using the recommended dose of NPK through 100% of biofertilizer made with microalgae *Mychonastes homosphaera* (formerly *Chlorella minutissima)* (Chlorophyta) showed the highest yield of leaf biomass compared to the control and the dose of mineral fertilizers for the spinach; for the baby corn, the yield with and without husk showed the best results with the biofertilizer; and for the cob length, it showed similar results to the chemical fertilizer [92].

#### 3.1.3. Hydrocarbon Microalgae-Based Biofertilizer

Hydrothermal carbonization, which converts biomass into hydrocarbon, is carried out at high pressures (above the water vapor pressure) and high temperatures (180–220 °C) in a liquid medium, for the treatment of biomass of microalgae, and it promises to be a good technique that provides excellent results in obtaining nutrients that can be found suspended in the liquid medium after the process, for example, nitrogen that is found mainly in the form of organic nitrogen, nitrates, and ammonium, or the orthophosphate that is formed from some polyphosphates present in the lignocellulosic raw material after the hydrolysis process [93,94,95]. In addition, biochar has been used as a biofertilizer or support material for the release of other fertilizers due to a series of positive effects it has on the environment, for example, higher profitability in the agricultural sector, restoration of degraded areas, and lower risk of eutrophication for the environment [96].

An increase in rice grain yield has been reported when fertilized with hydrocarbon from hydrothermal carbonization of *C. vulgaris* biomass thanks to the increase in ammonium ion NH_4_^+^ concentration in the soil after fertilization [86].

#### 3.1.4. Biofertilizer Enhanced with Microalgae 

Animal waste from cattle, pigs, and poultry generally has high nutrient values that are important for their use as fertilizer. These wastes cannot be directly treated with microalgae due to the presence of suspended solids and a high concentration of ammonium. Therefore, anaerobic digestion is carried out initially [97].

The bioavailability of nutrients has been studied when some corn digestates ensiled with dairy manure enriched with biomass of *Chlorella* sp. (10% of the total dry weight of the biofertilizer) in the growth of corn plants, giving greater value in the dry weight of the plant compared to the use of another genus of microalgae [98]. Suchithra et al. [99] also used the dry and macerated biomass of *Chlorella vulgaris* with cattle manure in the growth of tomato plants, giving excellent results, both in the useful life of the tomato, the size and in the concentration of nutrients, compared to the control (where the soil was not fertilized) and when *C. vulgaris* and cattle manure were applied separately. Another study was carried out with onion plants where growth, yield, leaf area, pigment content, and biochemical composition among other growth parameters were monitored, and it was found that the plant fertilized with cattle manure supplemented with *S. platensis* presented the highest growth factors analyzed, followed by the cattle manure supplemented with *C. vulgaris*, compared to the control [100]. A favorable result was obtained when the soil was treated with cow manure and two microalgae to analyze the growth of maize plants (length of leaf, root, dry, and fresh weight of the plant) for 75 days, as well as the content of macro and micronutrients and the yield of the plants, concluding that the best results were obtained when cow manure supplemented with *Arthrospira platensis* (formerly *Spirulina platensis;* Cyanobacteria)was used followed by the results obtained with cow + *C. vulgaris* [101].

## 4. Application Techniques of Microalgae-Based Biofertilizers

The simplest way to employ microalgae as a fertilizer is by directly applying the wet biomass in soil, but to make this practice viable, the energetic requirements must be optimized. Foliar fertilization with cellular extracts is also an alternative, and the results obtained from this technique are also profitable. A shorter time of flowering and in general growth are observed benefits of phytostimulating the crops with microalgae as a nutrient source [89,102].

In a study carried out by Castro et al. in 2020 [6], the environmental impact of a phosphate biofertilizer from microalgae against a commercial fertilizer (triple superphosphate), resulted in a higher value in all of the evaluated categories in the cycle life approach (fossil depletion, freshwater ecotoxicity, terrestrial ecotoxicity, particulate matter formation, freshwater eutrophication, terrestrial acidification, and climate change). In this sense, the selection and design of the processes for microalgae exploitation must be carefully studied. For that study, the process was a meat processing industry effluent, but to adopt a circular economy approach, the microalgae was added as a secondary treatment of wastewater and the production of biofuel, pigments, and fertilizer may present a more sustainable and economically feasible practice [103].

There are reports of growth-promoting factors (phytohormones) produced by microalgae, such as cytokinins, auxins, gibberellins, and abscisic acid, that are crucially involved in plant growth [89,102]. When these molecules are studied, it is most common to find the literature on microalgae extracts rather than crude biomass. In Figure 4, we summarized the challenges and opportunities of employing microalgae as fertilizer.

### Foliar and Soil Application of Microalgae Biomass 

Since algae can produce phytohormones, which have a direct impact on the physiological processes of plants, their ability to act as phytostimulants is also of particular interest and represents a significant advantage over conventional fertilizers. Phytohormones are organic substances that are generated in small quantities and control physiological processes in plants [104]. In addition to the endogenous hormones that plants produce, a variety of bacteria and fungi also manufacture and release phytohormones that have an impact on plant development [105]. The evidence of microalgal phytohormone synthesis is expanding rapidly. Numerous cyanobacteria species have been shown in various studies to be capable of producing auxins and cytokinins, which are essential for controlling nearly every aspect of plant physiology, including the architecture and growth of roots and shoots, as well as the development of vascular networks and organs [106]. Gibberellin has been discovered to be produced by cyanobacteria, such as *Arthrospira platensis*, and is involved in several biological activities, including stem lengthening, leaf expansion, early blooming, sex expression, fruit development, and the suppression of seed dormancy [107]. The presence of ethylene and abscisic acid (ABA) has been found in some strains according to several studies [102]. These phytohormones have a wide range of effects; ethylene has a direct impact on germination, flowering, senescence and abscission, acceleration of fruit ripening, and responses to biotic and abiotic stress; whereas ABA controls seed dormancy and serves as a significant stress hormone, particularly concerning drought stress and improvements in germination, plant root, shoot length and weight, leaf number, or flower parameters are the results of the production of these phytohormones [108]. Additionally, some reports mention increased levels of carbohydrates, proteins, pigments, nutrients, essential oils, or phytohormones in plant tissue, as well as improved resistance to abiotic stress. Most effects are favorable for plant development and encourage the use of microalgae as phytostimulants.

There is a broad variety of crops that have been tested to grow with microalgae extracts or directly biomass as an alternative to synthetic fertilizers. As the soil is affected by nutrient depletion even crop residues, manures, and mineral fertilizers result as inefficient alternatives to restore the nutrient richness of the soil. From this perspective, a biofertilizer plays a crucial role in soil recovery [109]. Li et al. [110] performed a pilot-scale microalgae production with a selenium-enriched culture where the raceway reactor configuration for microalgae growth also served as a domestic wastewater treatment [110]. *Phaseolus vulgaris* has been improved in selenium content and growth time. In this same year, Dineshkumar et al. evaluated the effect of *Chlorella vulgaris* on tomato crops (*Lycopersicon esculentum* Mill); the experimentation showed a positive result for plant growing, improving the height, number of stems, number of leaves, length, and root length at 75 and 100 % of *Chlorella vulgaris* dry biomass as a soil drench. In addition, the number of fruits and weight increased after the biofertilization procedure [109]. An onion crop (*Allium cepa*) was evaluated under *Spirulina platensis* and *Chlorella vulgaris* supplementation; in this study, besides the growth parameters that were in all the cases superior against the control, the anti-nutritional factors were evaluated and resulted in higher values for microalgae fertilized onion plants [100], highlighting another factor to keep in mind for microalgae-based fertilizer application.

In the practice of foliar supplementation, extracts, hydrolysates, and wet biomass of microalgae are common presentations [102,111,112]. A *Vigna mungo* seed was primed with *Spirulina platensis* extract, in this study, the main interest was not the growth increase but preventing the deterioration of the seed and improving the effective germination after aging [112]. Tejada-Ruiz et al. [111] evaluated the effect of a foliar hydrolysate from *Arthrospira platensis* (the content of cytokinins 2752, gibberellins 56.24, indoleacetic acid 10.3 ABA 1.03, salicylic acid 0.61, and jasmonic acid 0.84 ng/g) and silicon on the growth of *Pelagonium hortum*. In this study, the combination of potassium silicate improved the plant’s general growth, especially the number of flowers as a remarkable parameter since *P. hortum* is an ornamental plant.

Since the major target of microalgae soil enhancers is the synergism with a principal industrial process to make it feasible, the nutrient source (medium) and biomass harvesting/concentration methods are parameters that increase the costs of this application and its economic impact must be considered; however, in many cases, these parameters are not included as a critical step to optimize. Table 3 resumes some recent microalgae applications for crop improvement via foliar and soil microalgae supplementation.

It is documented that the production of molecules of hormonal nature that are identified in higher plants in microalgae (phytohormones) [115,116]. As in the previously mentioned study, the content of molecules, such as auxins, cytokines, gibberellins (GAs), abscisic acid (ABA), and others, are present in microalgae extracts in proportions of nanograms per gram [115,116]. These phytohormones may even be found in the residual methanol phase generated in the lipid extraction from microalgae. A recent study carried out by Pichler et al. [117] evaluated a total of eight phytohormones from lichen-forming microalgae photobionts. Interestingly from the six Trebouxiophyceae, the phytohormones were detected in the extracellular environment. Since the cellular lysis in the foliar crop’s microalgae treatment is a critical energetic or economic step, the extracellular production of these molecules opens the field for a more efficient implementation. However, simultaneously to experiments to maximize the obtention of phytohormones, the analytical methods must be adequate and explored to accurately estimate the physiological capacity of microalgae to produce these compounds intra- or extracellularly [118].

## 5. The Circular Economy of Waste to Microalgae-Based Biofertilizers

Microalgae hold immense potential for a range of applications, but their usage is currently restricted to lab conditions. Industrial-level applications have not advanced as much as they should for some reasons, the main one being the enormous economic expenses involved with large-scale applications. Two barriers to the production of raw materials for various uses by microalgae are the high cost of artificial media and the low biomass yield. One of the main advantages that microalgae-based biofertilizers have is that microalgae biomass can be obtained not only through conventional cultivation methods, but also through alternative, more sustainable ones. For this reason, one of the best approaches to address this issue is to cultivate microalgae from effluents, which has the potential to reduce process costs and produce microalgae biomass for a variety of uses [98]. This contributes to a more viable way that respects the principles of the circular economy and makes microalgae industrial production feasible due to the use of reusable resources as a culture medium. The circular bio-economy process will gain value from an integrated strategy that combines CO_2_ reduction, wastewater treatment, and biofertilizer generation to boost the sustainability and cost-effectiveness of microalgal farming [119].

Currently, 380 billion m^3^; of wastewater is produced annually across the world [120]. The basic inorganic/organic nutrients present in many wastewaters are ideal for the development of microalgae. Furthermore, it is possible to generate significant amounts of microalgal biomass in wastewater for the extraction of biomolecules for various uses due to the absorption of these nutrients present in wastewater streams that support rapid growth and biomass synthesis. [121]. Numerous studies have been conducted on the culture of microalgae and pollutant degradation/removal to handle various waste streams, such as municipal [122], industrial [123], and agricultural [124] effluents [125]. Some examples of these are presented in Table 4. Growing microalgae in wastewater have several benefits, including the oxidation of organic matter, treated water, and the breakdown of contaminants [126]. Additionally, it has been reported that the biomasses produced are rich in several bioproducts, including proteins, carbohydrates, pigments, and lipids that could be used in the chemical, pharmaceutical, and agricultural sectors [127].

Industrial effluent is a major cause of pollution in the environment. Wastewater from many industrial operations can serve as a nutrient-rich substrate for the development of microalgae [128]. Depending on the types of industrial processes and pollutants, many forms of industrial wastewater are produced [129]. One of the sectors that generate a large amount of wastewater is the food industry, and in most cases, the water used by this industry is of the highest quality [130]. Nonetheless, various factors should be considered to determine if the production of microalgae is economically feasible for the pure purpose of producing biofertilizers. Vázquez-Romero et al. carried out a techno-economic assessment of microalgae cultivation for their use as biorefineries, taking into account different operation, production, harvesting, and drying strategies [131]. After carrying out the economic analysis, it was determined that utilizing a single strain of microalgae year-round was shown to be the optimal course of action. For the harvesting process, it was estimated that the ultrafiltration technique was the one that raised the costs the least, in addition to implying a lower energy expenditure and having fewer biomass losses. It was found that spray drying was the most affordable method for drying biomass [132]. However, to cut expenses by up to 18%, the scale of production has to be increased to 10 hectares. The study also takes into account the usage of bulk fertilizers to improve the water that the microalgae would utilize as their growing medium [133]. Taking these data into account, the total costs for biomass production are close to 36.21 €/kg. However, the study also mentions that the use of effluents as a culture medium for microalgae could reduce the cost of microalgae by up to 67.50%, which would be equivalent to 11.76 €/kg, making it more competitive for the market. Therefore, the use of wastewater from the food industry as a culture medium for the growth of microalgae has great potential due to its organic matter content, richness in nutrients, biodegradability, and nontoxic characteristics allowing the cultivation of microalgal biomass at a cheap cost for biorefinery products in food industry effluent [134]. Although the cost of biofertilizers is still higher than that of conventional fertilizers indeed, an emphasis should also be given to the environmental benefits of these products as well as their roles in CO_2_ fixation and wastewater cleanup.

**Table 4 marinedrugs-21-00093-t004:** Comparison of different types of wastewaters used for the growth of microalgae species.

Microalgae	Type of Wastewater	Nutrient Removal Efficiency (%)	Biomass Yield (g/L)	Application	Composition	Reference
*Chlorella vulgaris*	Municipal wastewater	COD: 84.3% NO_3_-N: 82.62% NH_3_-N: 89% PO_4_^3−^-P: 85.15%	3.2	Bio-oil production	Lipids: 18.2% Protein: 55.24%	[135]
*Tetradesmus dimorphus* (formerly *Acutodesmus dimorphus*)	Dairy wastewater	COD: 91.71% NO_3_-N: 100% NH_3_-N: 100% PO_4_^3−^-P: 100%	0.84	Biofuel	Lipids: 25.05%, Protein: 38.69%	[136]
*Scenedesmus* sp.	Brewery wastewater	COD: 73.66% TN: 75.96% NH_3_-N: 89.99% TP: 95.71%	1.02	Wastewater purification	Chlorophyll: 20.40 mg/L Carotenoids: 7.54 mg/L Carbohydrates: 63.61 mg/L Lipids: 38 mg/L	[137]
*Chlorella vulgaris*	Clean in-place wastewater	COD: 75.0% NO_3_-N: 54.8% TP: 79.4%	-	Producing food-grade algae biomass	Lipid: 15.6% Proteins: 32%	[138]
*C. reinhardtii*	Dairy Wastewater	COD: 76% TN: 65% NH_3_-N: 65% PO_4_^3−^-P: 87%	1.14	Wastewater purification and lipid production	Lipids: 18.5%	[139]
*Auxenochlorella pyrenoidosa* (formerly *Chlorella pyrenoidosa*)	Soybean processing wastewater	COD: 77.8% TN: 88.8% NH_3_-N: 89.1% TP: 70.3%	0.64	Biomass cultivation	Lipids: 37%	[140]
*Chlorella vulgaris*	Slaughterhouse Wastewater	COD: 96.80% TN: 97.75% NH_3_-N: 57.74% TP: 56.66%	0.12	Wastewater purification	-	[141]
*Scenedesmus* sp.	Piggery Wastewater	NH_3_-N: 90% PO_4_^3−^-P 90% COD: 59%	0.054	Wastewater purification	-	[142]

## 6. Future Perspectives

The constant development in the research of microalgae-derived fertilizers brings with it a deep understanding of the various advantages, not only in terms of increasing crop productivity but also of the beneficial impact that this alternative can have on the environment and human health. The long-term use of biofertilizers would substantially decrease the use of chemical fertilizers, which, although they work to increase crop yield, also cause severe damage to the soil. On the other hand, the use of biomass as a biofertilizer does not contribute to soil degradation and at the same time, through its cultivation process, helps reduce CO_2_ emissions into the atmosphere. This duality presents a viable option to achieve sustainability goals in the industry and at the same time establish a feasible alternative to promote a circular bio-economy, which will significantly benefit agricultural practices, industry, and the general population.

Another advantage that microalgae have over conventional fertilizers is that they are living and modifiable organisms, meaning that their properties can be optimized to adapt to different types of crops and, in this manner, release substances that favor optimal growth or, on the other hand, release biopesticides, thus reducing losses and helping to control pests. In addition, by optimizing the balances of N, P, K, and other micronutrients, the biofertilizer will avoid generating an excess in the soil, and therefore prevent the pollution of water bodies.

It is estimated that the use of biofertilizers allows bioremediation and stops the loss of arable land, even allowing soil considered infertile to recover its properties inexpensively and more sustainably. To make microalgae usage a more feasible practice, further research efforts should be made toward the optimization of nutrient loads, culture systems, water reuse, industrial-scale harvesting techniques, and by-product extraction. These advances allow microalgae to adopt a greater number of applications that in one way or another have a positive impact on the environment, such as CO_2_ bio-capture systems and the United Nations and the almost 200 countries commitment to attend to the global warming effects due to climate change due to greenhouse gases increase; this could be a possible solution to the high demand of fertilizer needed for crop production for the global needs.

## 7. Conclusions

Microalgae biotechnology applications and biomass biorefinery emerge as a sustainable alternative for biofertilizer productions, taking advantage of biomass production from phytoremediation processes such as wastewater treatment and CO_2_ capture and impulse circular economy concept which represents an alternative to do profitable the phytoremediation process. Also, it can implement focus applications due to the properties of used microorganisms, where cyanobacteria and microalgae can promote differential phytostimulation activities, improving agricultural activities as alternative nutrient sources and growth improvers as promoting factors as phytohormones. The implementation of microalgae-based biotechnology represents a reduction in the carbon footprint in the industry and agriculture process.

## Figures and Tables

**Figure 1 marinedrugs-21-00093-f001:**
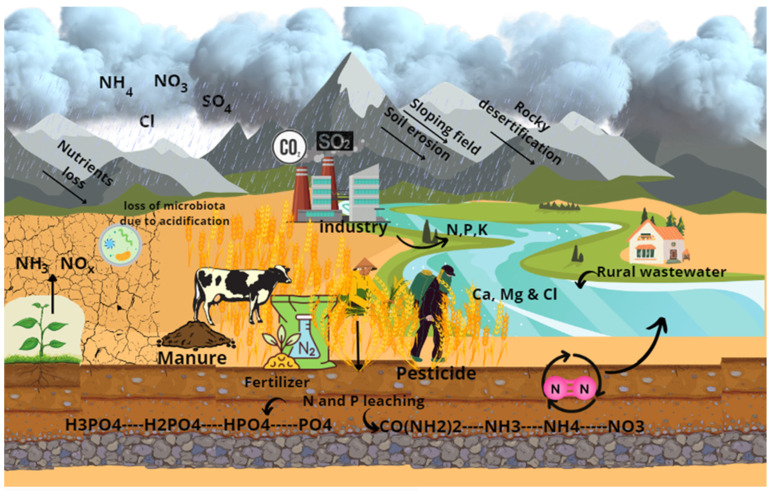
Effects of excess nutrients N, P & K in the soil.

**Figure 2 marinedrugs-21-00093-f002:**
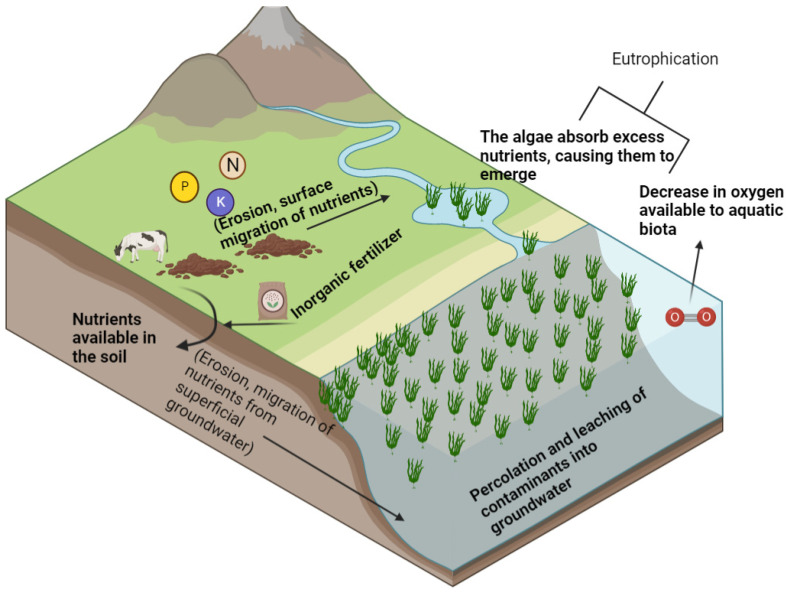
Leaching of nutrients (N, P & K) in bodies of water.

**Figure 3 marinedrugs-21-00093-f003:**
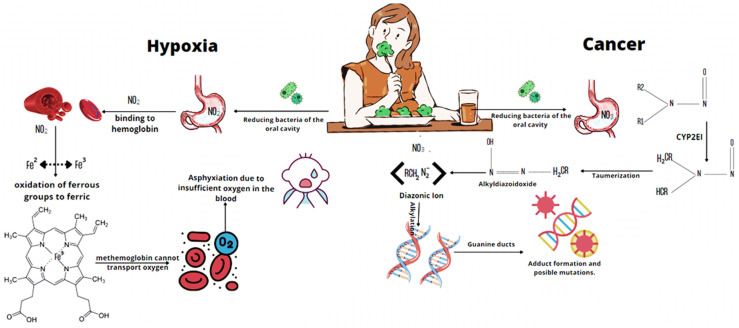
Nitrate metabolism in the body and related health problems.

**Figure 4 marinedrugs-21-00093-f004:**
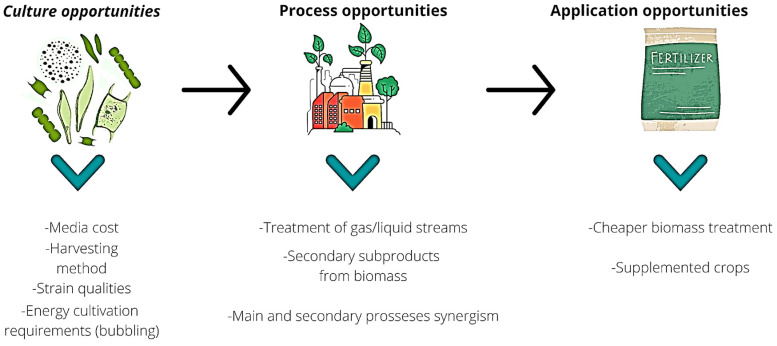
Microalgae biofertilizer challenges for process optimization/Optimization of microalgae fertilizer production.

**Table 1 marinedrugs-21-00093-t001:** Secondary micronutrients and their adequate concentration on soils for plant nutrition.

	Nutrient	Critical Concentration	Reference
Primary nutrients	Nitrogen (N)	25–50 mg/kg	[16]
	Phosphorus (P)	1 µM *	[17]
	Potassium (K)	141–370 mg/kg	[18]
Secondary nutrients	Calcium (Ca)	6–778 mg/kg *	[19]
	Sulfide (S)	>15 mg/kg	[20]
	Magnesium (Mg)	0.05–0.5% *	[21]
Micronutrients	Cobalt (Co)	15–25 mg/kg *	[22]
	Copper (Cu)	>0.04 mg/kg	[20]
	Boron (B)	>0.75 mg/kg	[20]
	Chlorine (Cl)	100 mg/kg *	[23]
	Iron (Fe)	>7.5 mg/kg	[20]
	Zinc (Zn)	>1.5 mg/kg	[20]
	Manganese (Mn)	>4mg/kg	[20]
	Molybdenum (Mo)	>0.2 mg/kg	[20]

* = Not adequate, only an average concentration in soil.

**Table 2 marinedrugs-21-00093-t002:** Comparison of some characteristics of the traditional fertilizers against the three types of biofertilizers.

Characteristics	Traditional Fertilizers	Biofertilizers		
		Bacteria	Fungi	Microalgae/Cyanobacteria
Environmental damage by degrading the soil, water contamination, and eutrophication induction.	✔	x	x	x
Creation of symbiotic bonds with the plant roots and microorganisms within the soil.	x	✔	✔	✔
Role in the nitrogen cycle making it available to the plant.	x	✔	✔	✔
Promotion of the solubilization of phosphorus.	x	✔	✔	✔
Soil fertility improvement.	x	✔	✔	✔
The slow rate of nutrient release for the consumption of the plant	x	✔	✔	✔
N fixation by individual strains, P solubilization, and hormone production for promoting the growth of the plant.	x	x	x	✔
CO_2_ capture and greenhouse emissions reduction capability during the addition of organic carbon to the soil.	x	x	x	✔
Industrial production and widespread used in the agriculture field.	✔	✔	✔	x

Table generated from the compilation of information from a variety of sources [7,25,76,77,78].

**Table 3 marinedrugs-21-00093-t003:** Foliar and soil microalgae application effects in plant crops.

Microalgae	Biomass Scale Production	Nutrient Source	Advantages	Disadvantages	Biomass Harvesting Method	Reference
*Chlorella* sp.	Pilot scale	Domestic Wastewater	*Phaseolus vulgaris* 3.2 times growth enhancement 3.5 times dry biomass	Low GP, GI, and SVI in high microalgae extract percentage. Potential contaminants carriage by the biomass from domestic water	Secondary clarifier/ sedimentation	[110]
*Chlorella vulgaris*	100 L	Conway medium	*Lycopersicon esculentum mill* shelf life increase in 2/3	Several energetic consuming steps to release intracellular content (freezing, microfluidization)	Filtration	[109]
*Arthrospira platensis* and *Chlorella vulgaris*	NM	NM	*Allium cepa* L. growth, yield, biochemical composition, and minerals improved	Anti-nutritional composition increase	NM	[100]
*Arthrospira platensis*	Laboratory	NM	*Vigna mungo* L. seed germination, speed germination, dry matter production, seedling length and biochemical composition improvements	Lower free sugar content	NM	[112]
*Arthrospira platensis*	100 m2 raceway reactor	Arnon medium	*Pelagonium hortum* L.H. Bailey flower number and FDW increase in saline environment, *Arthrospira platensis* and Si combination negative effect of NaCl content mitigation	NM	NM	[111]
*Arthrospira platensis* and *Scenedesmus* sp.	NM	NM	Petunia x hybrida increase in P foliar concentration	Enzymatic hydrolysis is required to obtain the hydrolysate	NM	[102]
*Chlorella vulgaris*	100 L	Conway medium	*Solanum lycopersicum* increased plant height, number of stem branches, number of leaves, leaves length, and root length	Ultrasound technology to produce cellular extracts not scalable	Filtration	[99]
*Chlorella* species (MACC-360 and MACC-38) and *Chlamydomonas reinhardtii*	Laboratory	Tris-acetate-phosphate media	*Medicago truncatula* increase in leaf area, early blooming for *Chlorella,* increase in biomass, pigments (chlorophylls and carotenoids) and flower number	Delayed blooming for *C. reinhardtii,*	Direct application	[113]
*Roholtiella* sp.	Laboratory	BG11	*Capsicum annuum* Increase of shoot length, root length, fresh weight, dry weight, spad index, number of leaves	NM	Centrifugation	[114]

NM = Not Mentioned, SS = Soil Supplementation, GP = Germination percentage, GI = Germination index, SVI = Seedling vigor index, FDW = Flower dry weight.

## Data Availability

Not applicable.

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
