# Peer review of "Microalgae-Based Biotechnology as Alternative Biofertilizers for Soil Enhancement and Carbon Footprint Reduction: Advantages and Implications"

_marinedrugs, 2023, doi:10.3390/md21020093_

Round 1
Reviewer 1 Report
The manuscript seeks an alternative source of fertilizer since conventional organic or inorganic fertilizers are unsuitable and negatively affect human health. It draws the critical line between the demand for alternative fertilizers and emerging microalgae cultivation for numerous purposes. However, several important issues shall be addressed before this manuscript is suitable for publication.
1. Phycostimulation activity can be explained as an extensive term. It can include the termination of the dormancy period or other physiological processes in the plant. However, the discussion throughout the review mainly points out the biofertilizer application. Specify the title or add more discussion.
2. Section 2 might be revised as it describes the negative impact of current technologies using conventional fertilizer. It represents human health impacts; it does not specify the effects on other organisms and the ecosystem.
3. Section 3 seems dangling. It describes microalgae-based fertilizer as dried biomass. However, the application of microalgae as living organisms that can produce other signaling molecules was not mentioned or discussed.
4. The comparison between current technologies of biofertilizer using fungi, bacteria, or other organisms shall be mentioned as it can show clear advantages and the gap that microalgae can fill to supply microalgae-based fertilizer. Consider comparing several parameters such as technology, economic values, etc.
5. Cyanobacteria are microalgae that can capture free N from the air and supply it to the plant. There is no significant review or discussion about these microalgae throughout the manuscript. Similarly, silica in diatoms may be taken into consideration as well. It is essential to discuss this in the subsection.
6. Supplying nutrients to the plant is the main objective of applying fertilizer. However, several other advantages, such as plant-microbes interaction, are also available in biofertilizers. Does algal-based fertilizer also possess this trait?
7. Other molecules, such as indole-3-acetic acid, have been reported to be produced in green microalgae. However, the amount of this molecule in algal biomass is relatively low since they only synthesize it for intracellular metabolisms. Please consider the fate of such molecules if the application method of microalgae as biofertilizers in modern agriculture.
8. Additional information in the table summarizing the current technologies of microalgae application as biofertilizers shall be added, such as advantages and disadvantages.
9. Ln 468 “Therefore, the use of wastewater from the food industry as a culture medium for the growth of microalgae has great potential due to its organic matter content, rich in nutrients, biodegradable, and nontoxic characteristics.” To utilize the wastewater as a substrate for algae to grow, it is crucial to compare the advantages for biomass application later in the process. Mainly, previous studies were developing algae as a biofuel. To reach the economic factors of biofuel and fertilizer, there might be more to discuss, such as cost of preparation, extraction method, the composition of biomass, and utilization of lipid-extracted biomass for other purposes such as application for fertilizer since it still contains high mineral contents.
10. Several minor errors must be avoided, such as Ln 52, N2O must be in correct writing format, etc.
Author Response
- Phycostimulation activity can be explained as an extensive term. It can include the termination of the dormancy period or other physiological processes in the plant. However, the discussion throughout the review mainly points out the biofertilizer application. Specify the title or add more discussion.
Response: The thistle of the article was fitted with the content
- Section 2 might be revised as it describes the negative impact of current technologies using conventional fertilizer. It represents human health impacts; it does not specify the effects on other organisms and the ecosystem.
Response: Section 2 deals with the damage to the flora and fauna of the ecosystems
- Section 3 seems dangling. It describes microalgae-based fertilizer as dried biomass. However, the application of microalgae as living organisms that can produce other signaling molecules was not mentioned or discussed.
Response: We took the advice and some parts of section 3 were rewrited. Also it was added a new section (3.1.1 Wet microalgae-based biofertilizers) to talk more about the effects of using living microalgae as a soil enhancer.
- The comparison between current technologies of biofertilizer using fungi, bacteria, or other organisms shall be mentioned as it can show clear advantages and the gap that microalgae can fill to supply microalgae-based fertilizer. Consider comparing several parameters such as technology, economic values, etc.
Response: The consideration to make a table for this comparison of different types of biofertilizers was made but the authors feel that this will take the review to another direction and our aim for the review is to analyze the reach and current applications of microalgae as alternative biofertilizers. Nonetheless we are open to discuss this comment further if the reviewers feel the same way after this round of corrections.
- Cyanobacteria are microalgae that can capture free N from the air and supply it to the plant. There is no significant review or discussion about these microalgae throughout the manuscript. Similarly, silica in diatoms may be taken into consideration as well. It is essential to discuss this in the subsection.
Response: We extended the discussion about this property and we added a paragraph of this characteristic in section 3.
- Supplying nutrients to the plant is the main objective of applying fertilizer. However, several other advantages, such as plant-microbes interaction, are also available in biofertilizers. Does algal-based fertilizer also possess this trait?
Response: The endosymbiotic interaction between microalgae, plants and bacteria was further explored in a paragraph addedn in section 3.1.
- Other molecules, such as indole-3-acetic acid, have been reported to be produced in green microalgae. However, the amount of this molecule in algal biomass is relatively low since they only synthesize it for intracellular metabolisms. Please consider the fate of such molecules if the application method of microalgae as biofertilizers in modern agriculture.
Response: Additional information was added after table 2 in order to clarify and exemplify the importance of these molecules. Thank you very much for your observation.
- Additional information in the table summarizing the current technologies of microalgae application as biofertilizers shall be added, such as advantages and disadvantages.
Response: The corresponding information as suggested was added in table 2
- Ln 468 “Therefore, the use of wastewater from the food industry as a culture medium for the growth of microalgae has great potential due to its organic matter content, rich in nutrients, biodegradable, and nontoxic characteristics.” To utilize the wastewater as a substrate for algae to grow, it is crucial to compare the advantages for biomass application later in the process. Mainly, previous studies were developing algae as a biofuel. To reach the economic factors of biofuel and fertilizer, there might be more to discuss, such as cost of preparation, extraction method, the composition of biomass, and utilization of lipid-extracted biomass for other purposes such as application for fertilizer since it still contains high mineral contents.
Response: The wastewater section is already expanded to address economic matters

Reviewer 2 Report
The manuscript summarized the adverse effects on the environment and human health caused by the improper use of chemical fertilizers. The fertilizer substitution potential of the emerging microalgae biofertilizer, the treatment and application methods of microalgae biomass, and the effects and potential of carbon reduction and wastewater treatment are also well summarized. This review will provide good insight for developing microalgae biofertilizer. There are some minor comments and suggestions for improvements:
1. The title does not well match the content of the text. The authors pay too much attention on the adverse effects of chemical fertilizer, and there are not too much new ideas in the statement of the side effects of chemical fertilizer. There is very limited in-depth statement on the soil remediation enhancement and carbon footprint reduction of microalgae biofertilizer in the manuscript.
2. It is suggested that the effects of microalgae biofertilizer should be summarized and analyzed in depth including the research data of soil physical and chemical properties and soil microbiome.
3. Compared with the nearly 200 million tons of chemical fertilizer used annually, the cost and yield of microalgae fertilizer will be the biggest bottleneck for chemical fertilizer replacement. The authors are encouraged to put forward their opinions on how to overcome the bottleneck and the implementation path.
Author Response
The manuscript was updated and your comments were attended
- The title does not well match the content of the text. The authors pay too much attention on the adverse effects of chemical fertilizer, and there are not too much new ideas in the statement of the side effects of chemical fertilizer. There is very limited in-depth statement on the soil remediation enhancement and carbon footprint reduction of microalgae biofertilizer in the manuscript.
Response: The title was adapted. Also with the global corrections the content was improved.
- It is suggested that the effects of microalgae biofertilizer should be summarized and analyzed in depth including the research data of soil physical and chemical properties and soil microbiome.
Response: The effects of microalgae-based fertilizers is analyzed more in-depth on section 3 and 3.1.
- Compared with the nearly 200 million tons of chemical fertilizer used annually, the cost and yield of microalgae fertilizer will be the biggest bottleneck for chemical fertilizer replacement. The authors are encouraged to put forward their opinions on how to overcome the bottleneck and the implementation path.
Response: The multiple microalgae biotechnological application for the bioremediation process could be a possible solution to satisfy the high demand for fertilizers in the world, this was included in the future perspective section.
I would like to thank your valuable comments. You can find the updated manuscript in attach.

Round 2
Reviewer 1 Report
The authors have addressed the comments very well. However, reviewer still suggests to add the discussion for other microbial comparison to show and strengthen the importance and special role of microalgae as biofertilizer. Additionally, the discussion of flora and fauna can still be strengthen by adding the biodiversity implications that reported from previous studies. References in table 1 can still be added to support the elements portions in plants nutrition.
Author Response
Comments and Suggestions for Authors
The authors have addressed the comments very well. However, reviewer still suggests to add the discussion for other microbial comparison to show and strengthen the importance and special role of microalgae as biofertilizer. Additionally, the discussion of flora and fauna can still be strengthen by adding the biodiversity implications that reported from previous studies. References in table 1 can still be added to support the elements portions in plants nutrition.
Response:
Table 1 was modified and detailed references were included. Also, the suggested discussion was added on page 6 previous to the 2.1 section and page 9 including Table 2.